# Defining Kinetic Properties of HIV-Specific CD8^+^ T-Cell Responses in Acute Infection

**DOI:** 10.3390/microorganisms7030069

**Published:** 2019-03-04

**Authors:** Yiding Yang, Vitaly V. Ganusov

**Affiliations:** 1Department of Microbiology, University of Tennessee, Knoxville, TN 37996, USA; yidingyang@gmail.com; 2National Institute for Mathematical and Biological Synthesis, University of Tennessee, Knoxville, TN 37996, USA; 3Department of Mathematics, University of Tennessee, Knoxville, TN 37996, USA

**Keywords:** acute HIV infection, vaccines, CD8^+^ T cells, immune response, multiple epitopes, competition, mathematical model

## Abstract

Multiple lines of evidence indicate that CD8${}^{+}$ T cells are important in the control of HIV-1 (HIV) replication. However, CD8${}^{+}$ T cells induced by natural infection cannot eliminate the virus or reduce viral loads to acceptably low levels in most infected individuals. Understanding the basic quantitative features of CD8${}^{+}$ T-cell responses induced during HIV infection may therefore inform us about the limits that HIV vaccines, which aim to induce protective CD8${}^{+}$ T-cell responses, must exceed. Using previously published experimental data from a cohort of HIV-infected individuals with sampling times from acute to chronic infection we defined the quantitative properties of CD8${}^{+}$ T-cell responses to the whole HIV proteome. In contrast with a commonly held view, we found that the relative number of HIV-specific CD8${}^{+}$ T-cell responses (response breadth) changed little over the course of infection (first 400 days post-infection), with moderate but statistically significant changes occurring only during the first 35 symptomatic days. This challenges the idea that a change in the T-cell response breadth over time is responsible for the slow speed of viral escape from CD8${}^{+}$ T cells in the chronic infection. The breadth of HIV-specific CD8${}^{+}$ T-cell responses was not correlated with the average viral load for our small cohort of patients. Metrics of relative immunodominance of HIV-specific CD8${}^{+}$ T-cell responses such as Shannon entropy or the Evenness index were also not significantly correlated with the average viral load. Our mathematical-model-driven analysis suggested extremely slow expansion kinetics for the majority of HIV-specific CD8${}^{+}$ T-cell responses and the presence of intra- and interclonal competition between multiple CD8${}^{+}$ T-cell responses; such competition may limit the magnitude of CD8${}^{+}$ T-cell responses, specific to different epitopes, and the overall number of T-cell responses induced by vaccination. Further understanding of mechanisms underlying interactions between the virus and virus-specific CD8${}^{+}$ T-cell response will be instrumental in determining which T-cell-based vaccines will induce T-cell responses providing durable protection against HIV infection.

## 1. Introduction

HIV-1 (HIV) remains a major global infectious disease with more than 35 million infected individuals, and millions of deaths due to AIDS every year [1,2]. Despite decades of research, a highly effective vaccine against HIV/AIDS is not yet available; several vaccine candidates failed in large phase II or III clinical trials [3,4,5]. One set of such failed trials investigated the efficacy of a CD8${}^{+}$ T-cell-based vaccine against HIV that had shown reasonable protection following the infection of immunized monkeys with SIV [6,7]. Although it is likely that multiple factors contributed to the failure of this vaccine in humans, the limited breadth and small magnitude of the vaccine-induced T-cell response might have been important [8,9]. However, the magnitude and breadth of HIV-specific CD8${}^{+}$ T-cell response needed for a protective vaccine are not well defined [9,10]. Although most recent vaccine developments have shifted toward the induction of broadly neutralizing antibodies [11,12,13,14], it is likely that the induction of both neutralizing antibodies and memory CD8${}^{+}$ T cells will be needed for adequate control of HIV [10,15].

Multiple lines of evidence suggest that CD8${}^{+}$ T cells play an important role in the control of HIV replication; some evidence is based on correlational studies in humans and some on experiments with SIV-infected monkeys [16,17,18]. In particular, (1) the appearance of CD8${}^{+}$ T-cell responses in the blood is correlated with a decline in viremia [16,19,20,21,22]; (2) the rate of disease progression of HIV-infected individuals is strongly dependent on MHC-I locus combinations [23,24,25]; (3) HIV escapes recognition from multiple CD8${}^{+}$ T-cell responses during the infection [16,26]. No consensus has been reached on the relationship between magnitude of HIV-specific CD8${}^{+}$ T-cell responses and viral load [27,28,29,30,31,32]; several studies, but not all, have indicated a statistically significant negative correlation between viral load and the number of Gag-specific CD8${}^{+}$ T-cell responses [32,33,34,35,36]. Important data also came from experiments on SIV-infected monkeys; depletion of CD8${}^{+}$ T cells prior to or after infection leads to significantly higher viral loads [37,38,39,40]. Some vaccination protocols in monkeys, in which high levels of SIV-specific CD8${}^{+}$ T cells were induced, resulted in a reduced viral load and, under certain conditions, apparent elimination of the virus [6,7,41,42,43,44].

Despite these promising experimental observations, following natural infection, CD8${}^{+}$ T-cell responses have not cleared HIV in any patient, or reduced viral loads to acceptably low levels in many individuals [16,45,46]. While some HIV-infected individuals do not appear to progress to AIDS and maintain high CD4${}^{+}$ T-cell counts in their peripheral blood (so-called long-term non-progressors or elite controllers, [46,47,48]), whether CD8${}^{+}$ T cells are solely responsible for such control remains undetermined [46,49,50,51,52,53]. It is clear that if we are to pursue the development of CD8${}^{+}$ T-cell-based vaccines against HIV, such vaccines must induce more effective CD8${}^{+}$ T-cell responses than those induced during natural HIV infection. However, the definition of a “more effective” response is not entirely clear. If induction of a broad (i.e., specific to multiple epitopes) and high magnitude CD8${}^{+}$ T-cell response is not feasible, it remains to be determined whether vaccination strategies should focus on the induction of broad and low magnitude or narrow and high magnitude CD8${}^{+}$ T-cell responses. The basic quantitative aspects of HIV-specific CD8${}^{+}$ T-cell responses induced during natural infection may indicate which parameters of vaccine-induced responses should be targeted for improvement so that the vaccine provides reasonable protection in humans.

There are several studies documenting the kinetics of HIV-specific CD8${}^{+}$ T-cell responses in humans from acute to chronic infection [54,55,56,57,58,59]. In some cases, the data are restricted to a few well-defined epitopes, often inducing immunodominant responses [59,60,61]. Similarly, only the kinetics of immunodominant CD8${}^{+}$ T cell responses to SIV in monkeys following vaccination have been analyzed and well quantified [62,63]. Many theoretical studies developed mathematical models of within-host HIV dynamics and their control by T-cell responses [64,65,66,67,68,69], but these models have not been well parametrized due to a lack of appropriate experimental data. Furthermore, these models involved different *a priori* assumptions on how CD8${}^{+}$ T-cell responses to HIV are generated and maintained; the dynamics of these responses are often responsible for the observed changes in viral load and kinetics of viral escape from T cells [64,68,70]. Further refinements of such models and investigations of the robustness of their predictions will benefit greatly from the systematic analysis of the kinetics of HIV-specific CD8${}^{+}$ T-cell responses. In particular, it remains unclear whether CD8${}^{+}$ T-cell responses specific to different epitopes of HIV compete during infection as many mathematical models assume [64,69,71]. Studies on the competition between CD8${}^{+}$ T cells specific to the same or different epitopes in mice are inconclusive, with some documenting competition and others a lack of competition [72,73,74,75,76,77,78,79,80,81,82]. A recent study using cross-sectional data suggested an absence of competition between CD8${}^{+}$ T-cell responses, specific to different HIV epitopes [83]. The absence of such interclonal competition would also predict that it is possible for a vaccine to generate a very broad HIV-specific CD8${}^{+}$ T-cell response.

In the present study, we performed mathematical-model-driven analysis of experimental data on viral load and HIV-specific CD8${}^{+}$ T-cell dynamics from a study of 22 patients who had been followed from acute to chronic infection [55]. The useful features of these data include the high temporal resolution of CD8${}^{+}$ T-cell responses and viral load measurement, with the detection of many viral epitopes recognized by CD8${}^{+}$ T cells using the ELISPOT assay. In contrast with several previous studies (e.g., [60,61,83]), which focused on a subset of well-defined epitopes and epitope-specific CD8${}^{+}$ T cells, we followed CD8${}^{+}$ T cell responses to the whole viral proteome, which enabled detailed quantitative investigation of CD8${}^{+}$ T-cell responses to HIV.

## 2. Material and Methods

### 2.1. Experimental Data

The data collection methods and patients’ details were described in detail previously [55]. Briefly, individuals with acute HIV-1 subtype B infection were recruited into the study, blood samples from the patients were taken at multiple, sequential time points over several months following symptomatic presentation. Patients were initially defined as HIV infected based on the presence of acute retroviral syndrome symptoms typically following a known/suspected recent high-risk HIV exposure incident [55]. All measurements were timed in days since onset of symptoms. The time interval between infection and onset of symptoms is likely to vary somewhat between individuals [84]. Viral load was recorded for all patients. Some patients started anti-retroviral therapy 3 years post-infection, but this was not used in the analysis since T-cell response measurements were done at most for 2 years. Patient MM38 started therapy within 100 days since symptoms and thus only data prior to treatment were included in the analysis. We did not have additional information on the age or sex of these patients. Viral sequences used to map HIV-specific CD8 T-cell responses were obtained within the 2–6 months since symptoms. Protein regions targeted by patients’ HIV-specific T-cell responses were mapped using peptide-stimulated interferon (IFNγ) ELISPOT assay or tetramer immunolabeling. Both assays show similar patterns of responses kinetics [55], but in our analyses we only used data obtained by ELISPOT. Please note that in patients WEAU, SUMA, and BORI, T-cell responses were not mapped to the whole proteome; in these patients, responses measured in a previous study [85] were followed over time. In total, there were data for 22 patients (two additional patients only had tetramer immunolabeling measurements and were therefore not included in the analysis). Experimental data on the dynamics of HIV-specific CD8${}^{+}$ T-cell responses and viral loads are shown in Appendix A. Original data can be requested from the corresponding author of the primary publication (Seph Borrow, persephone.borrow@ndm.ox.ac.uk).

### 2.2. Mathematical Model of CD8${}^{+}$ T-Cell Response to a Viral Infection

To quantify the kinetics of HIV-specific CD8${}^{+}$ T-cell responses, we used a simple $T_{\mathrm{on}}/T_{\mathrm{off}}$ mathematical model ([86], Figure 1). The model assumes that the response starts at time t=0 with frequency $E_{0}$ of epitope-specific CD8${}^{+}$ T cells that become activated at time $T_{\mathrm{on}}$. Activated T cells start proliferating at rate ρ and reach the peak at time $T_{\mathrm{off}}$. Thereafter, epitope-specific CD8${}^{+}$ T cells decline at rate α. The dynamics of the CD8${}^{+}$ T-cell response E(t) are therefore represented by the following differential equation:(1)$$\frac{dE}{dt}=\left\{\begin{array}{cc} 0, & \mathrm{if}\;t<T_{\mathrm{on}},\\ \rho E, & \mathrm{if}\;T_{\mathrm{on}}\leq t\leq T_{\mathrm{off}},\\ -\alpha E, & \mathrm{if}\;t>T_{\mathrm{off}} \end{array}\right.$$
with $E\left(0\right)=E_{0}$ as the predicted initial frequency of epitope-specific CD8${}^{+}$ T cells at time t=0 days since symptom onset.

Most immune responses (about 80%) had a detectable frequency at the first time point at which the response was measured, so we could not estimate when the response became activated ($T_{\mathrm{on}}$). Therefore, when fitting the mathematical model (Equation (Equation 1)) to such data, we set $T_{\mathrm{on}}=0$. This implies that we assumed each epitope-specific CD8${}^{+}$ T cell response is triggered at t=0 (onset of symptoms) with $E_{0}$ activated cells; this is clearly a simplification. In this way, the predicted initial frequency $E_{0}$ is a generalized recruitment parameter, which combines the true precursor frequency and the recruitment rate/time [86,88]. For a minority of responses (about 20%) there were one or several consecutive measurements in the first few days since symptom onset that did not result in detectable T-cell responses. In those cases, we set $T_{\mathrm{on}}$ as the first day with detectable measurements or the last consecutive day with non-detectable measurements. We fitted the model (Equation (Equation 1)) to the data on each measured epitope-specific CD8${}^{+}$ T-cell response in all patients using Mathematica 8 with nonlinear least squares by log-transforming the model predictions and data. For those responses that only expanded or only declined, we estimated only the expansion rate ρ or contraction rate α, respectively.

### 2.3. Statistics

Depending on the specific analysis, we used either parametric (e.g., Pearson correlation or linear regression) or nonparametric (Spearman’s rank correlation) methods. In most cases, significance was not strongly dependent on the method used and in cases when normality of the data was violated we used nonparametric tests. We used three metrics to estimate the strength of HIV-specific, Gag-specific, or Env-specific CD8${}^{+}$ T-cell response. Our focus on Gag and Env stems from previous observations on the relative importance of T-cell responses specific to these proteins in viral control [33,34].

The first metric was immune response breadth, which is the number of responses specific to either all HIV proteins, Gag, or Env at time *t*, n(t). For this metric, we took into account all time points at which CD8${}^{+}$ T-cell responses were measured for each patient. In some patients, there were missing measurements for some T-cell responses (marked “nd” for “not done”), so we tried two methods: (i) substituting “nd” with 1 (detection level); or (ii) removing that time point from the analysis. To estimate the breadth of the immune response it was important to exclude the data for that specific time point from the analysis; inclusion of such data might lead to an overestimation of the immune response breadth. There were subtle differences in estimated breadth using these two methods, but these did not substantially influence our conclusions. A second metric for the strength of the immune response was Shannon entropy (SE). While breadth only accounts for the number of responses, SE takes into account the relative abundance of individual responses, and reaches its maximum when all responses are of identical magnitude. SE at time *t* was calculated as $SE\left(t\right)=\sum_{i=1}^{n(t)}f_{i}\left(t\right)\log_{2}\left(f_{i}\left(t\right)\right)$ where n(t) is the number of HIV-, Gag-, or Env-specific T-cell responses at time *t*, and $f_{i}\left(t\right)$ is the frequency of the epitope-specific T-cell response in the total response at time *t*. Importantly, measurements of SE do not depend on “nd” or below-level-of-detection values; however, the number of detected responses n(t) may have a large impact on the actual value of SE. A third metric, Evenness index (EI) was calculated as the normalized SE: $EI\left(t\right)=SE\left(t\right)/\log_{2}\left(n\left(t\right)\right)$ where $\log_{2}\left(n\left(t\right)\right)$ is the maximum value SE can reach for n(t) immune responses. EI measures the degree of vertical immunodominance of HIV-specific T-cell responses [56] and varies between 0 and 1. Larger values indicate more “even” responses which, based on our and others’ previous work, should predict a longer time to viral escape from CD8${}^{+}$ T cell responses and therefore better virus control [56,89]. Both SE and EI are undefined for n=0. Furthermore, EI is ill-defined when only one immune response is measured per time point; this is relevant when looking at Gag- and Env-specific T-cell responses as some patients had few or none of those. We performed alternative analyses by (i) removing data points where n=1; or (ii) assigning EI=1 or EI=0 when n=1. These modifications did not influence most of our conclusions involving this metric. Because both viral load and breadth of T-cell responses changed within patients, in one set of analyses we calculated the mean breadth per time interval by averaging several measurements of breadth.

In addition to SE and EI, other measures of immunodominance could also be used. For example, Simpson’s diversity index is used in ecology to estimate species richness [90]. In our analyses, Simpson’s diversity index led to predictions similar to SE, so we have reported only the results for SE and EI here.

As some of our correlations turned out to be statistically nonsignificant we performed several power analyses to determine the numbers of patients needed to detect significance. We reanalyzed previously published data from Geldmacher et al. [34] to determine whether the small sample size in our cohort was responsible for the nonsignificant correlations. We performed these power analyses using a bootstrap approach by resampling from the data with replacement using $10^{3}$–$10^{4}$ simulations.

### 2.4. Ethics Statement

This paper uses experimental data obtained previously [55] and no new observations requiring patient consent or institutional review board approval have been performed.

### 2.5. Competing Interests Statement

None of the authors have any competing interests.

## 3. Results

### 3.1. Moderate Changes in the Breadth of HIV-Specific CD8${}^{+}$ T-Cell Response over the Course of Infection

While CD8${}^{+}$ T-cell responses are thought to play an important role in control of HIV replication, the kinetics of CD8${}^{+}$ T-cell responses specific to most HIV proteins, especially during the acute phase of infection, have not been quantified. Here, we reanalyzed data from a previous study that included patients infected with HIV-1 subtype B [55].

First, we investigated how many responses there were in a given patient and how the breadth of the HIV-specific CD8${}^{+}$ T-cell response changed over the course of infection. For every patient, we counted the maximum number of responses detected by ELISPOT assay to the whole viral proteome and their specificity (Figure 2). Similar to several previous studies [16,54,56], we found that most T-cell responses were directed against Gag and Env and this distribution changed little after 100 days since symptom onset (Figure 2A). Interestingly, responses to Nef, Integrase, or Reverse Transcriptase constituted a substantial fraction of all responses [91]. We found a median of eight epitope-specific CD8${}^{+}$ T-cell responses per patient, with two patients having over 15 responses and three patients having only three responses. Because of the potential limit of detection associated with ELISPOT assays, the true breadth of HIV-specific CD8${}^{+}$ T-cell response may be even higher [59]. The distribution of the number of responses in a given patient did not change significantly over the course of infection, except in patients with many responses in which some T-cell responses disappeared in chronic infection (Figure 2B and Appendix A). There was no change in the average total HIV-specific T-cell response over time in this cohort of patients (Appendix A).

The breadth of the CD8${}^{+}$ T-cell response, measured as the number of HIV-specific CD8${}^{+}$ T-cell responses (or breadth of protein-specific (such as Gag-specific) CD8${}^{+}$ T-cell responses) has been implicated in protection against disease progression [33,34,36,92]. Some, but not all, previous analyses suggested an increase in the breadth of HIV-specific CD8${}^{+}$ T-cell responses over time [54,55,93,94]. We found variable patterns for the change in breadth over time, i.e., there were patients with increasing breadth (e.g., patients MM45, MM48, MM49), decreasing breadth (e.g., MM43, MM55), or with non-monotonically changing breadth (e.g., MM23, MM42; Appendix A). Because there was no significant change in the average number of T-cell responses in all patients (Appendix A), we calculated the dynamics of normalized breadth for individual patients, dividing the number of HIV-specific T-cell responses detected at a particular time point in a given patient by the total number of responses in that patient (Figure 3). Our analysis suggested that there was a moderate but statistically significant increase in the average normalized breadth over time (from 85% to 95%), and this increase was limited to the first 35 days after symptom onset.

A relatively high breadth in the first month after infection, averaged over many patients (~85% of the maximum), may arise from the mixture of patients in the early and late stages of acute infection; it may be expected that patients with early acute infection have few CD8${}^{+}$ T-cell responses, whereas patients with late acute infection have many CD8${}^{+}$ T-cell responses. To address this caveat, we analyzed the dynamics of relative breadth in a subset of patients with a declining viral load, which may be an indication of early acute HIV infection (patients MM25, MM28, MM39, MM40, MM23, MM33, MM45, MM49, MM55, MM56). We found that similarly to the previous analysis, there was a statistically significant increase in the average (or median) relative breadth over time (ρ=0.36, p=0.004), and this increase was limited to the first 12 days after symptom onset. The average normalized breadth increased from 73% to 96% between 12 and 400 days after symptom onset. Together, our results suggest a moderate increase in T-cell response breadth by the first few weeks after symptom onset; however, there is a possibility that an increase in breadth may be larger for patients progressing from very early acute to chronic infection. In a recent paper [58] a moderate increase in CD8${}^{+}$ T-cell response breadth within the first several weeks of symptom onset and then relatively stable maintenance of breadth was observed in one of two patients; the second patient showed a large increase in CD8${}^{+}$ T-cell response breadth over time.

Although the immune response breadth is considered to be a good measure of effective immune response [10], there is no reason for this conjecture other than to simplify calculation. In fact, it is possible that many HIV-specific T-cell responses with small magnitudes do not contribute to viral control but would be counted when calculating immune response breadth. Studies in mice indicate that the efficacy of effector and memory CD8${}^{+}$ T cells in killing peptide-pulsed targets in the spleen is directly proportional to the T cell frequency [95], meaning responses with a low frequency would contribute little to the killing of targets. Other studies have suggested that equal magnitudes of T-cell responses may be beneficial by limiting viral escape [56,71]. Therefore, we introduced two additional measures of HIV-specific T-cell response efficacy, allowing us to quantify T-cell immunodominance (or richness): Shannon entropy (SE) and Evenness index (EI, see Materials and Methods for details). While SE has been used to measure HIV genome variability in sequence alignments, it has not previously been used to estimate immunodominance of immune responses. Our analysis suggested that both SE and EI increased over the course of infection and that this change was more significant for EI, in part because EI cannot exceed 1 by definition (Appendix A). However, the statistically significant increases in these two metrics were also mainly restricted to the first 40 days since symptom onset. Thus, the number and magnitude of evenness for HIV-specific CD8${}^{+}$ T-cell responses both appear to increase very early in infection and stabilize within 40 days of symptom onset.

### 3.2. Variable Correlations between Immune Response Breadth and Viral Load

Correlates of protection against disease progression of HIV-infected individuals are incompletely understood. It is well known that viral load is strongly correlated with risk of disease progression in HIV-infected patients [96] and many other parameters have been measured to reveal potential markers of protection. Among these, the breadth of HIV-specific CD8${}^{+}$ T-cell response has been widely emphasized as a potential predictor of viral control. Several studies found a statistically significant negative correlation between the number of Gag-specific CD8${}^{+}$ T-cell responses and viral load [33,34,36,92,97] whereas others did not [32]. In some of these studies, statistically significant negative correlations were based on relatively small numbers of patients, e.g., n=18 in Radebe et al. [92]. A negative correlation between viral load and breadth of Gag-specific CD8${}^{+}$ T-cell responses was also found using bioinformatic predictions of potential T-cell epitopes [35]. Negative correlations between viral load and CD8${}^{+}$ T-cell response breadth have generally been interpreted as an indication of protection even though it has been shown that viral load has an impact on the change in the number of Gag-specific T-cell responses over time [98].

We investigated the relationship between three different metrics of T-cell response efficacy: breadth, SE, and EI (see Material and Methods). For that, we calculated the average viral load and average metric for the whole observation period in a patient, during the acute (t≤100 days since symptoms) or chronic (t>100 days) phase of infection. None of the correlations between metric and viral load were significant, independent of the time period of infection or protein specificity (Figure 4 and Appendix A).

We also investigated whether changes in the immune response breadth over time were negatively correlated with viral load. Because there was a statistically significant increase in breadth within the first month of symptom onset, a negative correlation between the change in breadth and viral load may indicate that a larger breadth is associated with viral control. However, both negative and positive correlations were found in similar proportions, indicating that a greater breadth did not necessarily drive reduction in viral load (or vice versa). To determine if individual epitope-specific CD8${}^{+}$ T cells contribute to viral control, we calculated Spearman’s rank correlation coefficients between the magnitude of epitope-specific T-cell response and viral load for all T-cell responses over time (Figure 5). We found that there were disproportionally more negative than positive correlations, which suggested that increasing T-cell responses drive the decline in viral load (Figure 5A). By dividing the data into correlations during the immune response expansion ($t\leq t_{\mathrm{peak}}$, Figure 5B) and contraction phases ($t>t_{\mathrm{peak}}$, Figure 5C) we found that most negative correlations are observed when T-cell responses expand (and the viral load declines). These analyses are consistent with the idea that expansion of HIV-specific CD8${}^{+}$ T-cell responses is strongly associated with viral decline and that the contribution of T cells to viral control could be lower during chronic infection.

### 3.3. Most HIV-Specific CD8${}^{+}$ T-Cell Responses Expand Slowly and Peak Early

Several recent studies have quantified HIV dynamics during acute infection in patients either by using data from blood banks or by frequent sampling of individuals at high risk of HIV infection [84,99]. However, as far as we know there are no accurate estimates of parameters characterizing the kinetics of HIV-specific CD8${}^{+}$ T-cell response in acute infection. Therefore, we used a simple mathematical model (see Equation (Equation 1) in Material and Methods) to characterize the kinetics of epitope-specific CD8${}^{+}$ T-cell responses during acute HIV infection (Appendix A). Since our mathematical model (Equation (Equation 1)) describes T-cell responses specific to different viral epitopes in uncoupled form, all model parameters could be estimated for each T-cell response independently (Figure 6).

The dynamics of HIV-specific CD8${}^{+}$ T-cell responses were variable in individual patients. To further our analysis, we divided all HIV-specific T-cell responses into two subsets. In the first, a larger subset (about 80%) of T-cell responses were predicted to either expand or contract from the onset of symptoms (“early” responses, see Figure 1 and Figure 6). In a smaller subset, CD8${}^{+}$ T-cell responses had a delay $T_{\mathrm{on}}$ in the expansion kinetics (“delayed” or “late” responses, see Figure 1 and Figure 6).

Several parameter estimates differed between the two response subsets. In general, early responses expanded slower, peaked later, and had a higher predicted frequency $E_{0}$ than late responses (Figure 6). The average delay $T_{\mathrm{on}}$ in the expansion kinetics of late responses was only 15 days since symptom onset but some responses started expanding even later (Figure 6A). There was a minor difference in the timing of the T-cell response peak (Mann–Whitney, p=0.035) and over 90% of epitope-specific CD8${}^{+}$ T-cell responses peaked before 100 days since symptom onset (Figure 6B).

For the early responses, we found that there was an average of 97 antigen-specific CD8${}^{+}$ T cells per million peripheral blood mononuclear cells (PBMC) detected at the first time point (median, 13 IFN-$\gamma^{+}$ spot-forming cells (SFC) per million PBMC, Figure 6C). Please note that this is not very different from the experimental estimates of the frequency of human naive CD8${}^{+}$ T cells specific to viral epitopes [87,100]. To predict a theoretical frequency $E_{0}$ at which late responses would start to expand exponentially from t=0 days since symptom onset, we extended the fitted curve in the negative time direction to estimate the intercept with the *y*-axis. Around 24% of epitope-specific CD8${}^{+}$ T-cell responses including many “delayed” responses were predicted to have a precursor frequency $E_{0}<10^{-2}$ per million PBMC. Because this estimate is physiologically unreasonable [87,100], many of the “late” responses are likely to have started expanding after the onset of symptoms (i.e., were “delayed”).

Importantly, the majority (60%) of early epitope-specific CD8${}^{+}$ T cells expanded extremely slowly at a rate of <0.1 day${}^{-1}$ (median, 0.068 day${}^{-1}$, Figure 6D). An expansion rate of 0.1 day${}^{-1}$ corresponds to a doubling time of 7 days and this suggests that even in acute infection the majority of HIV-specific T cell responses expanded very slowly. In contrast, delayed responses expanded significantly faster, with a median rate ρ=0.31/day, which was only slightly lower than the T-cell expansion rate in response to the yellow fever virus vaccine [61]. A small fraction of early responses (6%) expanded at a fast rate of >0.5 day${}^{-1}$, but most responses contracted very slowly at a rate of <0.01 day${}^{-1}$ (Figure 6E). This implies that HIV-specific T-cell responses were relatively stable after their peak with a half-life of 70 days or longer. Thus, our analysis suggests that most HIV-specific CD8${}^{+}$ T-cell responses expand slowly, peak early, and remain relatively stable thereafter.

It was unclear why not all T-cell responses started expanding from symptom onset when viral loads are relatively high (Appendix A). For example, CD8${}^{+}$ T-cell responses to multiple epitopes of influenza virus or lymphocytic choriomeningitis virus (LCMV) in mice appear to start expanding almost simultaneously [81,88,101,102,103]. One hypothesis is that late T-cell responses are restricted to proteins that are not expressed at high levels during the HIV life cycle. However, this hypothesis was not supported by our data as delayed T-cell responses recognized multiple proteins, similarly to all T-cell responses in the cohort (Figure 2A and Figure 6F). A second explanation is that these delayed responses may be actively suppressed by the early responses. To investigate this, we calculated the Pearson correlation coefficients between 20 delayed responses (with a predicted frequency $E_{0}<0.01$, i.e., $T_{\mathrm{on}}>0$) and all other responses in these patients; most of these delayed responses were specific to Gag and found predominantly in patient SUMA0874 (Figure 2A and Figure 6F). Interestingly, only 20% of these correlations were negative, suggesting that other early responses continued expanding as late responses appeared. The observation that most early responses peaked after starting to expand further argues against an “active” suppression of delayed responses by early responses. Third, it is possible that late responses simply start from a smaller number of precursors [87]; this hypothesis could not be tested with our current data because estimated frequencies $E_{0}$ are unlikely to be true precursor frequencies. Fourth and finally, delayed expansion in the blood could simply be due to the retention of expanding T-cell populations in the lymphoid tissues. Testing this hypothesis would require measurements of HIV-specific T cell responses in the lymph nodes and/or spleen. Taken together, the reasons why some HIV-specific CD8${}^{+}$ T-cell responses appear late in the blood of infected patients remain unclear.

### 3.4. Evidence of Intraclonal Competition of CD8${}^{+}$ T Cells

Magnitude of epitope-specific CD8${}^{+}$ T-cell response is likely to be important in limiting virus replication (Figure 5). However, factors that influence the expansion kinetics of the CD8${}^{+}$ T-cell response and response peak in humans remain poorly defined. Recent work suggested that viral load in the blood of human volunteers during vaccination is the major determinant of the peak T-cell response following yellow fever virus vaccination [104]. We found that the frequency $E_{0}$ had a limited impact on the timing of the T-cell response peak (Figure 7A) and the rate of T-cell response expansion strongly affected the timing of the peak (Figure 7B). The latter suggests that more rapidly expanding responses peak early, which is markedly different from CD8${}^{+}$ T-cell responses in mice infected with LCMV where T-cell responses, specific to different viral epitopes, expand at different rates but peak at the same time [86,88,101,102].

Interestingly, we found that the expansion rate of epitope-specific T cell responses was strongly dependent on the average viral load during the expansion phase (Figure 7C) and on the estimated frequency $E_{0}$ (Figure 7D). The dependence of the expansion rate on viral load was nonlinear, in contrast with the linear or “saturating” function used in mathematical models describing the dependence of T-cell proliferation rate on viral load [64,66,71,86,94,105]. The observed decline in expansion rate of T-cell responses with a higher frequency $E_{0}$ strongly indicates the presence of intraclonal competition, suggesting that increasing precursor frequency of T cells by vaccination (an expected result of vaccination) may dramatically reduce expansion kinetics of such responses following exposure to HIV and this may limit T-cell efficacy in controlling virus replication. Similar intraclonal competition was also documented in some cases with T-cell responses in mice [106,107]. In particular, increasing the number of chicken ovalbumin-specific naive CD8${}^{+}$ T cells in mice reduced the expansion rate of the ovalbumin-specific CD8${}^{+}$ T-cell population following priming with ovalbumin [106].

Both the average viral load and predicted frequency $E_{0}$ had minimal impact on the peak CD8${}^{+}$ T-cell response (Figure 7E,F); interestingly, no correlation between CD8${}^{+}$ T-cell precursor frequency and peak T-cell response was found in mice [106]. The length of the expansion phase ($T_{\mathrm{off}}-T_{\mathrm{on}}$) had little influence on the peak immune response. It is, therefore, possible that the peak immune response was determined by virus-independent factors (e.g., cytokines); further analyses are needed to better understand the mechanisms limiting the magnitude of T-cell responses to HIV.

It has been previously proposed that some viral infections such as HIV and hepatitis C virus in humans and LCMV in mice induce a delayed CD8${}^{+}$ T-cell response, and this delayed response results in viral persistence [108,109]. We sought to determine whether HIV-specific CD8${}^{+}$ T-cell responses appear late in infection compared, for example, to viruses causing only acute infections in humans. It is clear that the expansion kinetics of virus-specific CD8${}^{+}$ T cell responses are likely to depend on viral load (e.g., Figure 7C,D). Therefore, for an appropriate comparison of acute and chronic viral infections we calculated the time intervals between the maximum observed viral load and the time when epitope-specific CD8${}^{+}$ T cells were predicted to reach their peak ($T_{\mathrm{off}}$). About 40% of HIV-specific T cells peaked only 10 days after the maximum viremia. A 10-day delay in CD8${}^{+}$ T-cell response peak after the peak viremia is similar to that which has been observed following yellow fever vaccination [60,61]. Therefore, these results suggest that many HIV-specific CD8${}^{+}$ T-cell responses are generated with similar kinetics relative to viral load for both acute and chronic infections in humans and yet most of them expand significantly slower than during an acute viral infection. This could, in part, be simply a consequence of HIV replication being slower than yellow fever virus replication.

### 3.5. Evidence of Interclonal Competition of CD8${}^{+}$ T Cells

Many mathematical models of the CD8${}^{+}$ T-cell response to HIV assume competition between responses specific to different viral epitopes [64,68,71]. In fact, the presence of such competition is important for explaining the kinetics and timing of viral escape from CD8${}^{+}$ T-cell responses [64,71]. However, to our knowledge, there is no experimental evidence of competition between different CD8${}^{+}$ T-cell responses in HIV infection. Studies of CD8${}^{+}$ T-cell responses to intracellular pathogens in mice reached conflicting conclusions, with some reporting no evidence for competition [73,78,101,110] and others reporting some evidence for competition [77,82,111,112]. A recent analysis of data on the magnitude of CD8${}^{+}$ T-cell responses specific to several HIV epitopes found no evidence for such interclonal competition during the chronic phase of HIV infection [83].

This previous study suffered from two major limitations: only a few CD8${}^{+}$ T-cell responses were analyzed, and the analysis was restricted to a single time point [83]. Therefore, we sought to determine if there is any evidence for competition between T-cell responses specific to different viral epitopes in the data of Turnbull et al. [55]. If there is competition between two responses, we expect that an increase in the magnitude of one response should lead to a decline in the magnitude of another, i.e., there should be a negative correlation between longitudinal changes in magnitudes of the two responses (Appendix A). We therefore calculated correlations between magnitudes of all pairs of epitope-specific CD8${}^{+}$ T-cell responses over time for every patient (Figure 8). The proportion of negative correlations indicating potentially competing immune responses varied by patient and was not strongly dependent on the time since infection (e.g., see Appendix A). In some patients, the proportion of positively and negatively correlated responses were similar (e.g., MM39, MM47, MM51) but in most patients, negative correlations were significantly under-represented as judged by the binomial test (Figure 8). Overall, approximately 18% of correlation coefficients were negative, suggesting that a small proportion of T-cell responses may be competing during the infection. However, in contrast with the assumptions of many mathematical models, the vast majority of responses do not appear to compete during the infection.

Previous analysis also suggested that in the presence of competition between epitope-specific CD8${}^{+}$ T cells, a larger number of responses should result in a smaller average size of epitope-specific T-cell response [83]. However, Fryer et al. [83] did not find a significant correlation between the number of responses and average size of epitope-specific T-cell response, indicating an absence of competition. One potential problem with this previous analysis was that it did not take viral load into account in the correlation, and it is possible that viral load may affect the strength of competition. For example, competition may be weak at high viral loads owing to an abundance of the antigen, and may be strong at lower viral loads (or vice versa). Furthermore, we showed that viral load influences the dynamics of HIV-specific CD8${}^{+}$ T-cell responses (Figure 7C) and thus may confound the correlation. Therefore, we repeated the analysis of Fryer et al. [83] by dividing the cohort data into three groups with different average viral loads (low, intermediate, and high, Figure 9). During the acute infection (t≤100 days after symptom onset) there was a statistically significant negative correlation between the number of responses and the number of T-cell responses (Figure 9C) suggesting interclonal competition. However, significant negative correlations were not observed for all time periods or all viral loads (e.g., t>100 with low or high viral load, Appendix A); thus, overall, by correcting for multiple comparisons we must conclude that there is no correlation between T-cell response breadth and average size of epitope-specific T-cell response. The two types of analyses (longitudinal in Figure 8 and cross-sectional in Figure 9) may thus have different power in detecting competition between immune responses. Our analysis of longitudinal data suggests that a sizable proportion of HIV-specific T-cell responses may be competing during infection.

## 4. Discussion

It is generally accepted that CD8${}^{+}$ T cells play an important role in controlling HIV replication. Features of HIV-specific CD8${}^{+}$ T cell responses that are important in mediating this control remain incompletely understood. T-cell specificity, polyfunctionality, and ability to proliferate have been cited as important correlates of protection [17,33,47,113]. Here, we analyzed the kinetics of the CD8${}^{+}$ T-cell response to the whole HIV proteome in patients controlling HIV poorly (which are the majority of HIV-infected individuals), and thus identified features associated with poor viral control.

In these patients, HIV infection induced a reasonably large number of CD8${}^{+}$ T-cell responses, most of which were generated during the earliest stages of infection (first 35 days after symptom onset). On average, CD8${}^{+}$ T-cell response breadth increased moderately during the first month since symptom onset and remained relatively stable for the next year. However, breadth varied differently in individual patients. In some patients, breadth increased two-fold over the course of 2 months after symptom onset, and in some patients, breadth remained constant or even declined. Importantly, a minimal change in CD8${}^{+}$ T-cell response breadth from symptom onset to chronic phase was also observed in three patients from the Center of HIV Vaccine Immunology cohort [94]. However, our finding seems to contradict a conclusion reached by Turnbull et al. [55] who found that the median breadth of CD8${}^{+}$ T-cell response increased from 2 to 6. The major difference between our analysis and the previous study is how we counted responses. Turnbull et al. [55] only counted responses that peaked within 2–3 weeks post symptoms, whereas we counted all detected responses.

Because of the high variability in the rate of exponential growth of CD8${}^{+}$ T-cell responses (e.g., Figure 7B) it is perhaps expected that only a few rapidly expanding responses should be observed early in infection. Later in infection, immune response with slower expansion rates would be detected, creating the impression of T-cell response breadth increasing with time. This idealistic interpretation may be an artifact of a limited sensitivity of ELISPOT and difficulty tracking T-cell response at the place of their generation, i.e., secondary lymphoid tissues. Better methods of T cell response detection in the blood and tissues are likely to provide a more complete picture of the dynamics of T-cell response breadth.

Because T cell response breadth may not be stable over the course of infection in individual patients, interpreting relationships between the breadth and other parameters, e.g., viral load, must be done with care. For example, it was observed that a change in the number of Gag-specific CD8${}^{+}$ T-cell responses with time was dependent on a patient’s viral load, suggesting that a larger breadth in chronic HIV infection may be the consequence and not the cause of a lower viral load [98].

We found no significant correlation between breadth, SE, or EI of HIV-, Gag-, or Env-specific CD8${}^{+}$ T-cell responses and viral load. This was in contrast with several (but not all) previous studies that identified a statistically significant negative correlation between the number of Gag-specific T cell responses and viral load [33,34,36,92]; some of those studies included patient cohorts of a similar size. This could be due to limited power in our study. Power analysis indicated that for a sufficiently large number of patients, statistically significant correlations could be found; however, such correlations were dependent on the measure of immune response efficiency. Efficiency measured as the number of Gag-specific T-cell responses was negatively correlated with viral load, whereas EI for HIV- or Gag-specific T-cell responses was positively correlated with viral load. The latter result, if confirmed in a larger cohort, is surprising, since T-cell responses of a similar magnitude were predicted to limit viral escape from T cells [56,94], and would therefore be expected to lead to a lower viral load.

It is not clear whether the small number of patients in our cohort (n=22) was responsible for the absence of a statistically significant correlation. Two previous studies also involved a relatively small number of patients and yet reached a statistically significant negative correlation between the number of Gag-specific T-cell responses and viral load [34,92]. Statistically significant results may arise in underpowered studies by chance [114], and a small number of patients in the study by Radebe et al. [92] may indicate an accidental statistically significant correlation. To investigate the potential difference between our result and that from Geldmacher et al. [34], we reanalyzed the data from the latter (data were provided by Chriss Geldmacher). The re-analysis revealed several major differences between our study and theirs. First, we found that Geldmacher et al. [34] detected more Gag-specific responses than Env-specific responses (slope of the Env vs. Gag regression was 0.11 with $p\ll 10^{-3}$ when compared to slope = 1; *t* test). In Turnbull et al. [55] data, the number of Gag and Env-specific T-cell responses were more similar (slope = 0.56, p=0.07 for the comparison with slope = 1; *t* test). Second, the correlation strength between the number of Gag-specific T-cell responses and viral load was previously overestimated; a nonparametric Spearman’s rank correlation test resulted in a higher, but still significant, *p* value (p=0.013) than that found previously (see Figure 2 in [34]; the published value was p=0.0016). Third and finally, we found that the statistical significance of the negative correlation was driven exclusively by four patients (out of 54) with many Gag-specific responses (≥6); removing these patients from the analysis made the correlation between viral load and number of Gag-specific CD8${}^{+}$ T-cell responses statistically nonsignificant (p=0.085). Resampling data from 18–22 patients from the Geldmacher et al. [34] cohort with replacement demonstrated low power in correlation between T-cell response breadth and viral load (power =46%); however, including the four outliers with high numbers of Gag-specific T-cell responses increased the power to 63%. Together, these results suggest that the potential protection by Gag-specific T-cell responses may not extend to all Gag-specific T-cell responses and may be a feature of only some patients. This interpretation is consistent with previous analyses that only looked at T-cell responses to defined Gag epitopes, and not to the whole gene [33,97]. More studies are needed to understand the protective nature of Gag- specific CD8${}^{+}$ T-cell responses; for example, the breadth of Gag-specific CD8${}^{+}$ T-cell responses did not predict the control of HIV after cessation of antiretroviral therapy in patients treated for acute HIV infection [115].

An additional important part of our analysis is an illustration of other metrics that can be used to evaluate the potential efficacy of CD8${}^{+}$ T-cell responses such as SE and EI. While it is clear they can complement a commonly used measure of efficacy (response breadth), these metrics have a strong limitation in that they ignore data from patients with no immune response, and EI is ill-defined for cases when only one immune response is present. Furthermore, calculation of these metrics requires measurement of the magnitude of epitope-specific T-cell responses.

By fitting a simple mathematical model to the longitudinal dynamics data for epitope-specific CD8${}^{+}$ T-cell responses, we estimated the parameters for T-cell responses in HIV infection. We predict that the vast majority of HIV-specific T cell responses (80%) recognize HIV early and expand (or are already contracting) during the onset of symptoms. These T-cell responses expanded extremely slowly, at a rate of <0.1 day${}^{-1}$, indicating that vaccines may need to induce responses with significantly quicker expansion kinetics. A small proportion of responses (20%) had a delayed expansion, and these late responses expanded at significantly higher rates than early responses. All responses appeared to be relatively stable after reaching their peak (the contraction rate was <0.01 day${}^{-1}$ for most epitope-specific CD8${}^{+}$ T cells).

Slow expansion of the early T-cell responses may be due to intraclonal competition for resources such as antigens. Indeed, we found a strong negative correlation between the predicted initial frequency of the response and the rate of response expansion, which is consistent with the presence of intraclonal competition. Several previous reports documented the presence of such competition under some, often unphysiological, circumstances (e.g., by artificially increasing the number of naive CD8${}^{+}$ T cells specific to an antigen) [106,107,116]. Slow expansion of T-cell responses may also arise as an artifact of the measurement of T-cell response magnitude as frequency (i.e., number of spots per million PBMC); however, because most of our total responses reach only about 1% of PBMCs (e.g., Figure 7E) and in general, about 10% of PBMCs are CD8${}^{+}$ T cells (personal communication from Seph Borrow), this alternative seems unlikely. The presence of intraclonal competition may strongly limit the magnitude of epitope-specific T-cell responses induced by vaccination.

A previous study found that the amount of yellow fever virus in the blood of volunteers greatly affects the magnitude of CD8${}^{+}$ T-cell response induced by vaccination [104]. In our analysis, however, this correlation was not significant if we corrected for multiple comparisons (Figure 7F). More work is needed to understand the factors regulating the magnitude of the T cell response following acute and chronic viral infections, as these may be different.

If broad HIV- or Gag-specific CD8${}^{+}$ T-cell responses are protective (as several studies have suggested; see above), induction of a broad T cell response may be difficult in the presence of interclonal competition. One previous study suggested that interclonal competition between CD8${}^{+}$ T-cell responses specific to different viral epitopes is absent in chronic HIV infection [83]. Interestingly, we found that the vast majority of HIV-specific T-cell responses (about 82%) appeared to have “synchronous” dynamics. Yet a substantial fraction of all responses did show evidence of competition when an increase in the magnitude of one response was associated with a decline in another (Figure 8). The relative fraction of such potentially “competing” T-cell responses varied by patient. Interestingly, using the method of Fryer et al. [83] to correlate the average size and number of T-cell responses did not allow the detection of competition. This indicates that longitudinal data may provide a higher power for detecting competition between epitope-specific CD8${}^{+}$ T-cell responses. Our results thus suggest that interclonal competition may potentially limit the breadth of vaccine-induced CD8${}^{+}$ T-cell responses.

It should be emphasized, however, that correlation does not necessarily indicate causality and negative associations between kinetics of individual T cell responses may arise for reasons unrelated to competition. Understanding why some responses are discordant while others increase or decrease in unison is likely to shed more light on the degree of T-cell competition during HIV infection. Recent work suggests that competition between HIV-specific CD8${}^{+}$ T cells for access to infected cells may influence the rate of virus escape [71,94]. Detecting competition in a biological system is a complicated problem (e.g., [117]). Direct fitting of classical mathematical models (Lotka–Volterra and predator–prey) revealed that these models can be consistent with some data but in some cases failed to accurately describe the data. Therefore, using mathematical models alone does not allow discrimination between alternative mechanisms of T-cell response competition, and further experiments are needed. One possible way of investigating whether responses compete is to boost the magnitude of a given response (e.g., by therapeutic vaccination) and see if this influences the magnitude of other T-cell responses. Clinical evidence suggests there is limited competition between humoral immune responses specific to different infections [118].

Several important caveats could not be addressed in this study. These include issues with experimental data and mathematical model assumptions. First, CD8${}^{+}$ T-cell responses were mapped at 2–6 months after symptom onset, so some T-cell responses appearing earlier or later than that time point could have been missed in the analysis. It is important to note, though, that mapping of CD8${}^{+}$ T-cell responses is often done at a single time point (e.g., [54,56,58]), meaning such analyses suffer from a similar limitation. Second, the IFNγ ELISPOT may not be sensitive enough to detect all the responses, and some evidence suggests that the sensitivity of this method may vary during the infection [59]. This is likely to affect some parameters but not others; for example, estimates of the rate of expansion of HIV-specific CD8${}^{+}$ T cell responses are likely to be dependent on ELISPOT sensitivity. Third, responses were measured only in the blood whereas interactions between the virus and T cells are likely to occur in lymphoid tissues. This problem is unlikely to be resolved in human studies because it will be difficult to obtain longitudinal samples of lymphoid tissues from patients. Fourth, the simple $T_{\mathrm{on}}/T_{\mathrm{off}}$ model may not fully describe T-cell response kinetics, especially during early acute infection. However, this model has been successful in describing the dynamics of the CD8${}^{+}$ T-cell response to viral infections in both mice and humans [61,86,88,119]. Fifth, averaging of the viral load to infer correlations between parameters may not be fully appropriate because in many patients there were large changes in viral load over time (Appendix A). However, explicit inclusion of viral load dynamics in some simple models proved difficult. Sixth, the data do not include the virus ramp-up phase, meaning the earliest CD8${}^{+}$ T-cell responses may be missed. Indeed, this might be an issue with many of the recent analyses and, to date, the available data on CD8${}^{+}$ T-cell response during the virus expansion phase are limited. It should be noted that the data in which viral load in the blood is measured soon after exposure (e.g., [84]) often comes from individuals who are at high risk of acquiring HIV infection, and thus virus dynamics in such patients may not represent an “average” patient. Seventh, alignment of patient’s data by the day since symptom onset may be misleading as different patients are likely to experience symptoms at different times after infection. Methods such as Fiebig staging or Poisson fitter may allow better alignment data in terms of days since infection [120,121] but the accuracy of these novel methods has not been well studied, in part, because the exact date for HIV infection is rarely known. Finally, a small number of patients (n=22) and limited information on background of the patients may limit general applicability of our results to other populations of HIV-infected individuals.

In summary, our study provides basic information on the kinetics of CD8${}^{+}$ T-cell responses specific to the whole HIV proteome given the limitations of current methods of measuring such responses in humans. Understanding the complex underlying biology of interactions between the virus and virus-specific CD8${}^{+}$ T-cell response, and of the factors driving changes in T cells, is instrumental in determining which T-cell-based vaccines induce a T-cell response exceeding that induced during natural HIV infection. We expect that such vaccines alone would induce responses with a substantial impact on virus replication. Results of the present analysis will also be helpful in developing better calibrated mathematical models of T-cell responses to HIV, which will be valuable in predicting whether and how T-cell-based vaccines can provide protection upon infection with the virus [10,122].

## Figures and Tables

**Figure 1 microorganisms-07-00069-f001:**
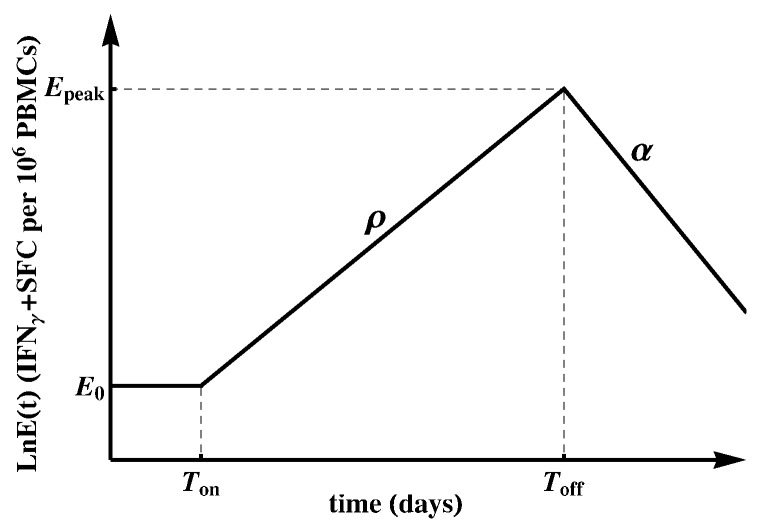
Schematic representation of the $T_{\mathrm{on}}/T_{\mathrm{off}}$ mathematical model fitted to the epitope-specific CD8${}^{+}$ T-cell response kinetics data [86]. In this model, $E_{0}$ epitope-specific naive CD8${}^{+}$ T cells become activated at time $t=T_{\mathrm{on}}$ and start proliferating at rate ρ. At $t=T_{\mathrm{off}}$, T-cell response peaks and declines at rate α. We refer to $E_{0}$ as the predicted initial frequency of epitope-specific CD8${}^{+}$ T cells [87]. Evidently, $E_{0}$ may over- or under-estimate the response precursor frequency depending on exactly when the T cells became activated and how adequate the mathematical model is for describing immune response data during the expansion phase.

**Figure 2 microorganisms-07-00069-f002:**
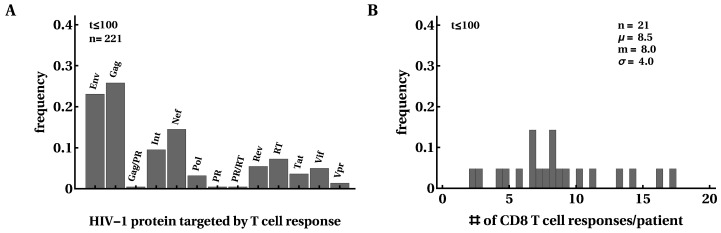
Most HIV proteins were recognized by CD8${}^{+}$ T-cell responses. We calculated the frequency at which HIV proteins were recognized by CD8${}^{+}$ T cells; overall, 50% of responses were directed against Env or Gag (**A**). m=8 CD8${}^{+}$ T cell responses were detected in this cohort of 22 patients at any given time point after infection (**B**). In B (and other figures in the paper), μ denotes the average, *m* is the median, and σ is the standard deviation. The distributions are shown for the first 100 days after symptom onset but, overall, distributions changed little over the course of 400 days of infection. Patient SUMA0874 was excluded from the analysis in B due to a lack of measurements of all T-cell responses at all time points.

**Figure 3 microorganisms-07-00069-f003:**
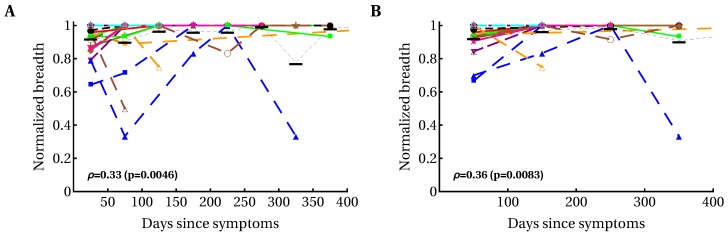
Modest yet statistically significant increase in the average normalized T-cell response breadth over the course of the first year of HIV infection. We divided the observations into different time bins ((**A**) 50-day intervals; (**B**) 100-day intervals) and calculated the relative breadth for the corresponding interval. The relative breadth was calculated as the number of HIV-specific CD8${}^{+}$ T-cell responses detected in a given time period divided by the number of all responses measured for that patient in all time periods; data were averaged to simplify presentation. Averaging did not influence the statistical significance of conclusions. Colors and symbols represent the data from different patients as shown in Appendix A. Black horizontal bars denote the mean relative breadth for that time interval for all patients. There was a statistically significant increase in relative breadth (Spearman’s rank correlation coefficient ρ and *p* values indicated on panels). There was no change in the average total immune response in all patients (Appendix A). Detailed analysis of the relative number of CD8${}^{+}$ T-cell responses in individual patients revealed variable patterns: constant breadth, increasing breadth, decreasing breadth, and breadth changing non-monotonically over time (Appendix A). Also, no overall change in the average breadth (un-normalized) was observed (Appendix A). We observed a similarly modest but significant increase in SE and EI of HIV-specific CD8${}^{+}$ T-cell response with time (Appendix A).

**Figure 4 microorganisms-07-00069-f004:**
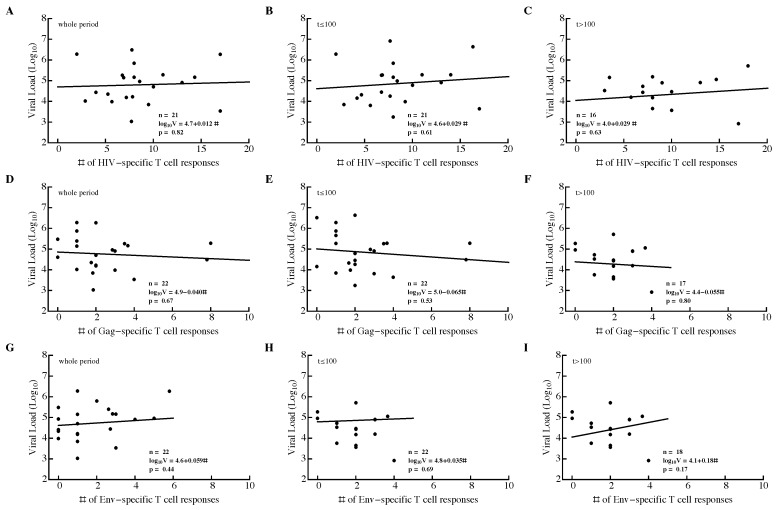
Breadth of HIV-specific CD8${}^{+}$ T-cell response in a patient does not correlate significantly with average viral load. We calculated the average number of HIV-specific (**A**–**C**), Gag-specific (**D**–**F**), and Env-specific (**G**–**I**) CD8${}^{+}$ T-cell responses over the whole observation period (**A**,**D**,**G**), during acute infection (t≤100 days since symptom onset; (**B**,**E**,**H**)), or during chronic infection (t>100 days since symptom onset; (**C**,**F**,**I**)) and $\log_{10}$ average viral load in that time period. The average viral load during infection was not dependent on the breadth of the Gag-specific CD8${}^{+}$ T-cell response during the infection (**D**–**F**). Patient SUMA0874 was excluded from the analysis in (**A**–**C**) due to insufficient measurements of all T-cell responses at all time points.

**Figure 5 microorganisms-07-00069-f005:**
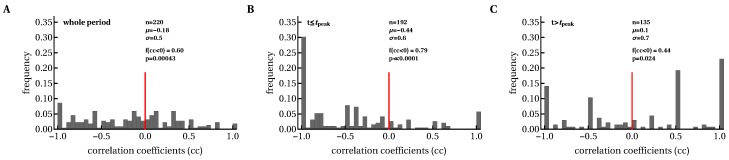
Expanding CD8${}^{+}$ T-cell responses were negatively correlated with viral load before T-cell numbers reached their peak values. We calculated Spearman’s correlation coefficients between longitudinal changes in viral load and epitope-specific CD8${}^{+}$ T-cell responses in each patient during the whole period (**A**), and before (**B**) and after (**C**) the peak of CD8${}^{+}$ T-cell response. The f(cc<0) value denotes the fraction of negative correlation coefficients (cc), and *p* values are indicated for the binomial test of equal distribution of positive and negative correlations.

**Figure 6 microorganisms-07-00069-f006:**
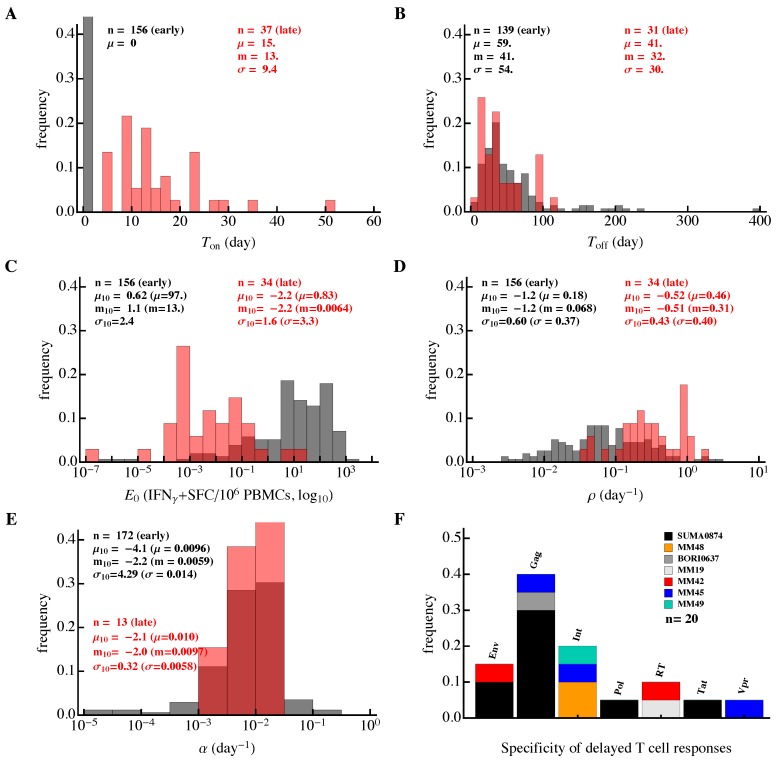
Differences in the kinetics of early and late HIV-specific CD8${}^{+}$ T-cell responses. We fitted the $T_{\mathrm{on}}/T_{\mathrm{off}}$ model (Equation (Equation 1)) to the data on the dynamics of epitope-specific CD8${}^{+}$ T-cell response in each patient and plotted the distribution of the estimated parameters. The results are presented separately for T cell responses that started expanding (or contracting) from the first observation (“early” responses, about 80% of all responses; black) or delayed responses, which were undetectable at one or several initial time points (“late” responses; red). Panels show distributions for (**A**) time of expansion of T-cell response ($T_{\mathrm{on}}$), (**B**) time to peak of each T-cell response ($T_{\mathrm{off}}$), (**C**) initial predicted frequency of epitope-specific CD8${}^{+}$ T cells ($E_{0}$), (**D**,**E**) expansion (ρ) and contraction (α) rates of T-cell responses, respectively, and (**F**) proteins recognized by late CD8${}^{+}$ T-cell responses. In (**A**–**E**), *n* represents the number of fitted responses, and μ, *m* and σ represent mean, median and standard deviation, respectively ($\mu_{10}$, $m_{10}$, and $\sigma_{10}$ are mean, median, and standard deviation for $\log_{10}$-scaled parameters). Late responses were predicted to have a higher expansion rate ρ (Mann–Whitney, p<0.001) and smaller frequency $E_{0}$ (Mann–Whitney, p<0.001) than early responses.

**Figure 7 microorganisms-07-00069-f007:**
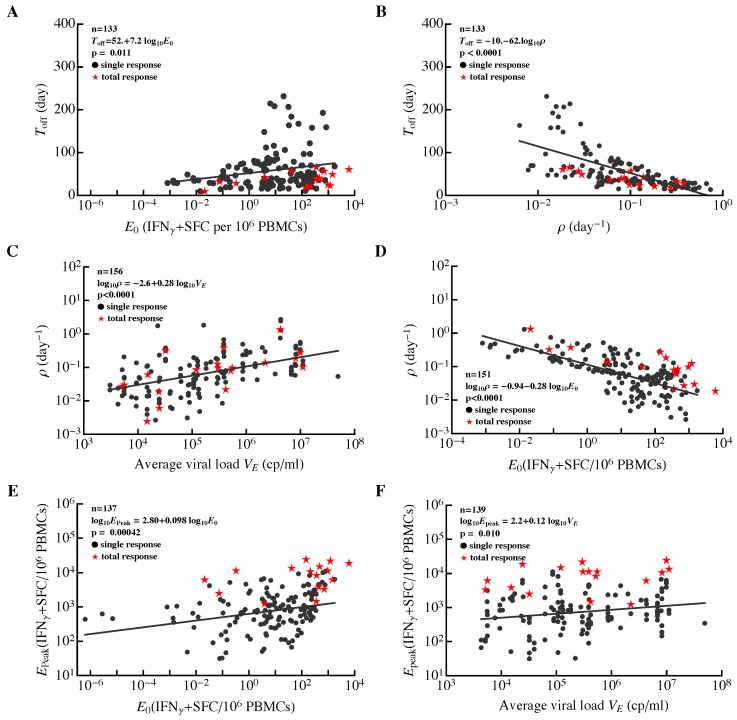
Correlations between major parameters determining dynamics of HIV-specific CD8${}^{+}$ T-cell responses in acute infection. For all epitope-specific CD8${}^{+}$ T-cell responses in all 22 patients (circles) or the total HIV-specific CD8${}^{+}$ T-cell response per patient (stars), we estimated the initial frequency of epitope-specific CD8${}^{+}$ T cells ($E_{0}$), rate of expansion of T-cell populations (ρ), time of the peak of the response ($T_{\mathrm{off}}$), rate of contraction of the immune response after the peak (α), predicted peak values reached by the epitope-specific CD8${}^{+}$ T-cell response ($E_{\mathrm{peak}}=E\left(T_{\mathrm{off}}\right)$), and the average viral load ($V_{E}$). Solid lines denote regression lines; regression equations and *p* values are indicated on individual panels for all epitope-specific CD8${}^{+}$ T-cell responses. The total HIV-specific CD8${}^{+}$ T-cell response showed a similar trend to all epitope-specific CD8${}^{+}$ T-cell responses. Panels show correlations between the timing of the immune response peak $T_{\mathrm{off}}$ and predicted frequency $E_{0}$ (**A**), $T_{\mathrm{off}}$ and ρ (**B**), expansion rate ρ and average viral load $V_{E}$ (**C**), ρ and $E_{0}$ (**D**), peak immune response $E_{\mathrm{peak}}$ and $E_{0}$ (**E**), and $E_{\mathrm{peak}}$ and $V_{E}$ (**F**). For a given patient, we calculated the total HIV-specific CD8${}^{+}$ T-cell response as the sum of all epitope-specific CD8${}^{+}$ T-cell responses at the same time point (i.e., by ignoring “nd”). For patient MM42, we could not fit the $T_{\mathrm{on}}/T_{\mathrm{off}}$ model to the dynamics of total CD8${}^{+}$ T cell response data because of wide oscillations in the data. Identified relationships did not change if estimates for responses with unphysiological initial frequencies ($E_{0}\leq 10^{-2}$) were excluded from the analysis.

**Figure 8 microorganisms-07-00069-f008:**
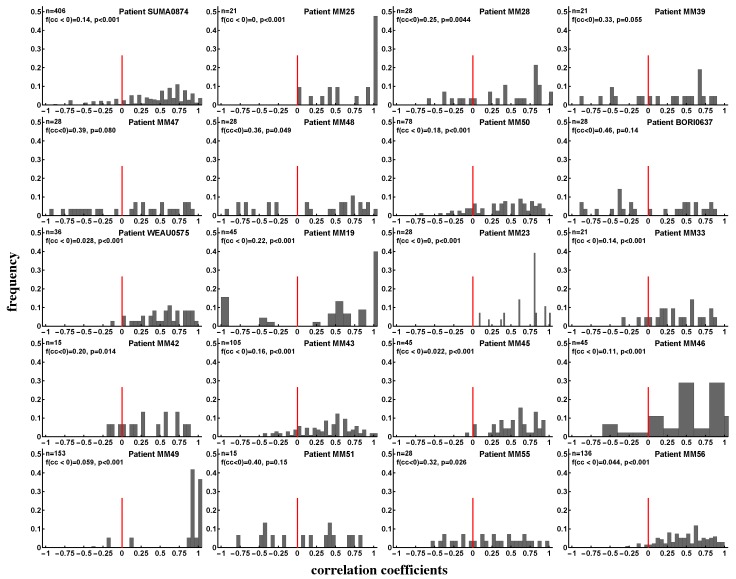
Evidence of interclonal competition between epitope-specific CD8${}^{+}$ T cell responses. We calculated Spearman’s rank correlation coefficients between longitudinal changes of pairs of epitope-specific CD8${}^{+}$ T cell responses in a given patient (see individual panels) and plotted the distribution of these coefficients. Panels show the number of correlations (*n*), fraction of negative correlation coefficients (f(cc)<0), and *p* values for the deviance of the distribution from uniform, found using the binomial test with null being the equal fraction of positive and negative correlations. We found that the majority of CD8${}^{+}$ T-cell populations expand and contract in unison and therefore do not appear to compete during the infection. Overall, discordant dynamics (negative correlation coefficients) were observed for 18% of all responses irrespective of the stage of infection (acute or chronic). Patients MM38 and MM40 were excluded from the analysis for having too few correlation pairs (two or three).

**Figure 9 microorganisms-07-00069-f009:**
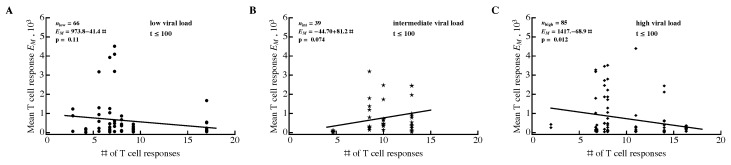
Average size of epitope-specific CD8${}^{+}$ T-cell response is unrelated to the number of HIV-specific T-cell responses. For every patient, we calculated the average number of HIV-specific CD8${}^{+}$ T-cell responses and the average density of epitope-specific T cells in a given observation period. To exclude the contribution of viral load to this relationship, we divided all 22 patients into three groups according to their mean viral load (low $\log_{10}$ viral load: 3.40–4.44 (disks) (**A**); intermediate viral load: 4.60–5.03 (stars) (**B**); high viral load: 5.25–6.83 (diamonds) (**C**)). Groups were estimated using the Manhattan Distance with the FindClusters function in Mathematica. Regression lines and corresponding *p* values are indicated on individual panels. Overall, results varied by time period and most correlations were not statistically significant (Appendix A).

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
