# Peer review of "Defining Kinetic Properties of HIV-Specific CD8+ T-Cell Responses in Acute Infection"

_microorganisms, 2019, doi:10.3390/microorganisms7030069_

Round 1
Reviewer 1 Report
The manuscript by Yang and Ganusov analyzed previously published data and reports the kinetics of CD8+ T-cell response to the HIV proteome. The study furthers our knowledge regarding the magnitude of CD8+ T-cells that may be required to be elicited by a vaccine regimen to exert a protective response. While the cohort size was small the analysis the study is relevant from vaccine point of view. However, increasing the number of samples analyzed would have added extra strength to the manuscript. The differences reported between the prior study and in this study are expected due to variations in the methods used thus pointing the need to develop well calibrated universal mathematical models that can help in evaluating/predicting the protection efficacy of a T-cell based vaccine. The study is straightforward and the discussion of the results is thorough and informative. The manuscript is very well written.
Author Response
Response. Having more samples would be indeed fantastic but unfortunately that particular study (by Turnbull et al.) had only data from these patients and we do not have access to other data. We have added a note in the discussion highlighting this limitation of our analysis.
Reviewer 2 Report
Defining kinetic properties of HIV specific CD8+ T-cell responses in acute infection, Yang and Ganusov
General Comments
This is a well written paper which addresses important questions of the role of CD8+ T-cell responses to control of HIV infection. Strengths of the current study include the use of the whole viral proteome to measure breath over time after infection and implementation of a simple mathematical model based on Shannon Entropy (SE) or Evenness Index (EI) to measure the breadth of epitope specific CD8 T cell responseS timed in days since onset of symptoms. The authors provide an excellent review of literature and discuss the relevance of their work in context of current understanding of the role of CD+ T-cell responses in HIV infection. The results show that CD8+ T-cell response is mounted very early after onset of symptoms in HIV infection and increase very slowly, while later responses expand at higher rates. Once activated, the CD8+ T-cell responses remain stable for long periods of time after peak expression. Intraclonal competition may potentially limit breath of CD8+ T-cell responses during infection.
Specific comments
1. It is not clear from the methods as to how acute infection was defined. Symptoms alone tend to be non-specific and are not a good indicator of HIV infection. The authors mention that viral load data is available, and may provide some differentiation between early and late infection, and that a few individuals with sequential viral loads declining over time provide evidence of early infection. Were results of serology such as the quantitative HIV Ag/Ab Combo Architect or BioPlex assays or Western Blot assays that may be useful in defining relative time of infection, available? Is there any other evidence of time of infection for the other patients?
2. Were any of the individual in this study under antiviral therapy, since that would certainly skew the results, and should be noted.
3. On page 16, the authors state that this population controls HIV poorly. That applies to majority of HIV infected patients, since Elite Controllers and long term non-progressors occur quite infrequently. Is there any additional information available about the population studies, such as risk factors (MSM, women, IVD) and demographics (male, female, age) since that may also be relevant to CD8 responses?
4. The CD8+ T-cell responses were measured within 6-months after symptoms which is actually a pretty broad time period and well after time of acute infection. The results of Fig 3 grouped by 50- and 100-day intervals do not address events in early infection (<30 days), when CD8+ response is expected to be most active. The statistically significant increase in the average normalized breadth over time (85%-95%) as seen in Fig 3 is due to small subset of patients who have lower breadth at the first time point and increase by time of next measure. Most individuals are already at maximal breadth by the time of the first test. If these include individuals who had been infected for several months at the time of first ELISPOT, the mature CD8 response by that time may skew the results of what is really happening in acute infection.
5. The major conclusion that CD8+ responses are relatively stable with no overall significant change in breath over >1 yr seems justified. The three individuals who show a decline in breadth at subsequent points. Could something else be going on in these individuals, such as starting of therapy or other infections? Simply averaging these in may not give a good picture of what is going on.
Author Response
Specific comments
1. It is not clear from the methods as to how acute infection was defined. Symptoms alone tend to be non-specific and are not a good indicator of HIV infection. The authors mention that viral load data is available, and may provide some differentiation between early and late infection, and that a few individuals with sequential viral loads declining over time provide evidence of early infection. Were results of serology such as the quantitative HIV Ag/Ab Combo Architect or BioPlex assays or Western Blot assays that may be useful in defining relative time of infection, available? Is there any other evidence of time of infection for the other patients?
Response. There are several alternative ways to define if a patient experiences acute HIV infection. At the time the study was conducted (in 2007-09), definition was based by admitting physicians based on “acute retroviral syndrome” (per Turnbull et al. description). There was no other possibility to “align” the patients using Fiebig staging or using more sophisticated tools such as Poisson Fitter from LANL HIV database. Yet, all the analyzed patients had a declining viremia from the first samples indicating (but not proving) acute infection.
2. Were any of the individual in this study under antiviral therapy, since that would certainly skew the results, and should be noted.
Response. Thank you for this comment! Our data indicate that few patients started ART but only one patient (MM38) started it within 100 days since symptoms; other patients started therapy 3 or more years since onset of symptoms. The data for MM38 were only included until the therapy started. We mention this detail in our Materials and Methods section now.
3. On page 16, the authors state that this population controls HIV poorly. That applies to majority of HIV infected patients, since Elite Controllers and long term non-progressors occur quite infrequently. Is there any additional information available about the population studies, such as risk factors (MSM, women, IVD) and demographics (male, female, age) since that may also be relevant to CD8 responses?
Response. Given the viral load dynamics it is true that patients in our cohort fall into the majority of HIV-infected patients. Unfortunately, we did not have more detailed information on individuals sex, although we suspect that vast majority was male and MSM. We also did not have information on other details (age, HLA) in that particular study. Therefore, there may be biases associated with the small number of patients analyzed. We now state this explicitly in the paper.
4. The CD8+ T-cell responses were measured within 6-months after symptoms which is actually a pretty broad time period and well after time of acute infection. The results of Fig 3 grouped by 50- and 100-day intervals do not address events in early infection (<30 days), when CD8+ response is expected to be most active. The statistically significant increase in the average normalized breadth over time (85%-95%) as seen in Fig 3 is due to small subset of patients who have lower breadth at the first time point and increase by time of next measure. Most individuals are already at maximal breadth by the time of the first test. If these include individuals who had been infected for several months at the time of first ELISPOT, the mature CD8 response by that time may skew the results of what is really happening in acute infection.
Response. Perhaps our phrasing was not precise. Immune responses were measured from the onset of symptoms which is in acute phase of infection. However, to measure CD8 T cell response there is a need to have viral sequence to generate proteins and peptides to measure CD8 T cell response by ELISPOT. HIV does evolve over time, so the original authors (Turnbull et al.) generated full length HIV genomes using samples from 6 months samples (in some patients) or at earlier time points for other patients (see Figure S7 for specific regions where HIV genomes were sequences for mapping T cell responses). We now state this more explicitly.
5. The major conclusion that CD8+ responses are relatively stable with no overall significant change in breath over >1 yr seems justified. The three individuals who show a decline in breadth at subsequent points. Could something else be going on in these individuals, such as starting of therapy or other infections? Simply averaging these in may not give a good picture of what is going on.
Response. None of these three patients started therapy in the time frame analyzed, so this decline in breadth is not due to this specific cause. We don’t have any other additional information on why these individuals had reduced breadth at later time points. We agree that averaging breadth is perhaps simplistic but we further state that breadth dynamics does vary by individual, and no change in breadth over time is observed in the majority of patients. Our main point, though, is that change in breadth is not large, even if there is an increase or decline, change in breadth takes long time. In part, this is to address previous predictions that kinetics of HIV escape from T cells may be driven by change in T cell response breadth. Here we show that change in breadth with time is too small to account for such an effect.